# The Integral Role of Tight Junction Proteins in the Repair of Injured Intestinal Epithelium

**DOI:** 10.3390/ijms21030972

**Published:** 2020-02-01

**Authors:** Zachary M. Slifer, Anthony T. Blikslager

**Affiliations:** Department of Clinical Sciences, College of Veterinary Medicine, North Carolina State University, Raleigh, NC 27607, USA; zmslifer@ncsu.edu

**Keywords:** barrier function, tight junction, repair, occludin, claudin, NHE2, ClC-2

## Abstract

The intestinal epithelial monolayer forms a transcellular and paracellular barrier that separates luminal contents from the interstitium. The paracellular barrier consists of a highly organized complex of intercellular junctions that is primarily regulated by apical tight junction proteins and tight junction-associated proteins. This homeostatic barrier can be lost through a multitude of injurious events that cause the disruption of the tight junction complex. Acute repair after injury leading to the reestablishment of the tight junction barrier is crucial for the return of both barrier function as well as other cellular functions, including water regulation and nutrient absorption. This review provides an overview of the tight junction complex components and how they link to other plasmalemmal proteins, such as ion channels and transporters, to induce tight junction closure during repair of acute injury. Understanding the components of interepithelial tight junctions and the mechanisms of tight junction regulation after injury is crucial for developing future therapeutic targets for patients experiencing dysregulated intestinal permeability.

## 1. Intestinal Epithelium as a Selective Barrier

The intestine is lined with a monolayer of columnar epithelium that maintains two critical functions: (1) selectively filtering luminal contents, including nutrients, water and electrolytes, to allow for their translocation into the circulation and (2) forming a barrier to prevent the translocation of luminal toxins, commensal or pathogenic microorganisms, and foreign antigens into the circulation [1]. Under homeostatic conditions, these functions are regulated by both transcellular and paracellular pathways, the latter of which are primarily maintained by apical tight junction proteins through paracellular pore and leak permeability pathways [2,3]. The paracellular pathway is associated with the charge and size selective transport of materials through the space between intestinal epithelial cells.

Intestinal barrier homeostasis is disrupted through tight junction protein dysregulation, which occurs via a variety of injurious events, including microbial degradation and bacterial toxin exposure, exposure to cytotoxic agents, exposure to pro-inflammatory cytokines such as IFNγ and TNFα, intestinal autoimmune disease such as Celiac disease, and intestinal ischemia [4,5,6,7,8,9]. The loss of tight junction integrity results in the formation of a third pathway, known as the high-capacity and nonselective unrestricted permeability pathway, which can allow for the unrestricted movement of microorganisms and large proteins across the paracellular space [10]. An inability to rapidly repair the tight junctions in order to restore epithelial barrier function is detrimental to the patient, as it can result in various pathologies, including sepsis and multiple organ dysfunction [11,12]. Therefore, understanding factors that can regulate the tight junction complex during repair of injured intestinal epithelium is crucial for developing future therapeutic targets.

## 2. Tight Junction Protein Structure

Tight junctions are made up of a number of protein elements, including transmembrane claudins (total of 27 mammalian claudins), as well as myelin and lymphocyte (MAL) and related proteins for vesicle trafficking and membrane link (MARVEL) [13,14,15,16,17,18]. MARVEL domain-containing proteins are a component of a larger group of tight junction-associated MARVEL proteins (TAMPS) that include transmembrane proteins such as occludin and tricellulin [18,19,20]. Other tight junction-associated transmembrane molecules include junctional adhesion molecules (JAM-1, -2, and -3) that can regulate the formation of tight junctions and migration of neutrophils [21,22,23,24,25,26]. Additionally, intracellular scaffold proteins such as zonula occludens (ZO) -1, -2, and -3 play an integral role in tight junction protein assembly and link tight junction transmembrane proteins with the actin cytoskeleton [27,28,29,30].

Intestinal claudins exist in two different classes: sealing claudins and pore-forming claudins [31]. Increased membrane expression of sealing proteins results in a ‘tighter’ epithelial barrier, further restricting the movement of luminal contents through the paracellular space. Sealing tight junction proteins include claudins-1, -3, -4, -5, -8, -11, -14, 18, and -19 [17]. Alternatively, increased membrane expression of pore-forming proteins (including claudins-2, -10a/-10b, -15, -16, and -17) reduces the selectivity for luminal contents that can pass between epithelial cells, thereby increasing paracellular permeability [17]. Each pore-forming claudin has an ionic charge specificity for cations or anions as well as ionic size selectivity, thereby increasing the permeability for ions based on claudin-specific permeability characteristics. An interesting interaction between the two classes of claudins has been observed with the displacement of pore-forming claudins by sealing claudins. For example, in a claudin-8 transfected Madin–Darby canine kidney II (MDCK II) cell line in which claudin-8 expression occurred in the absence of doxycycline, claudin-8 displacement of claudin-2 was visualized upon immunofluorescent imaging [32,33]. Specifically, claudin-8 served to replace claudin-2 in tight junction strands in this model, which reduced the number of functional paracellular cation pores [32].

Tight junction protein expression in the intestinal tract is tissue- and age-specific. For example, claudin-2 is normally expressed in the human small intestine but is reported to be expressed only in the colonic crypt of fetal humans and absent in the adult colon under homeostatic conditions [34]. Overall, claudin-2 generally exhibits higher expression in leaky epithelial tissues, such as colonic tissues from a subset of patients with inflammatory bowel disease [35]. Additionally, its protein expression was detected throughout the crypt–villus axis of human small intestines but was only detected in undifferentiated crypt cells of human colonic tissue [36]. Other claudins follow suit regarding crypt–luminal axis expression with some pore-forming claudins (-2, -10, -13, -15) being restricted to the intestinal crypt base in murine tissue while other sealing claudins (-3, -4, -7, -8) are expressed in luminal epithelial cells [37,38,39,40].

### Special Functions of Select Tight Junction Proteins

As discussed throughout the remainder of this review, regulation of tight junction proteins is a vital component of epithelial barrier repair after injury. However, specific tight junction proteins can have additional special functions that are species and age dependent. In the case of claudin-4, there is an age-dependent disparity between cellular localization of tight junction proteins [41]. Intestinal porcine claudin-4 is localized to the apical surface of porcine jejunal enterocytes within the first two days of age and only localizes to the lateral surface between adjacent epithelial cells beyond two days of age [41]. This age-specificity of claudin-4 localization in piglet jejunum may be due to immunological naivety that newborn piglets experience. Piglets must be able to acquire and absorb immunoglobulins and other immune-related molecules, including cytokines and antimicrobial peptides, from colostrum within the first day of life to prevent death via bacterial sepsis [42]. It is reported that this age-specific, apical surface localization of claudin-4 occurs in jejunal enterocytes at the same period of time in which the vital immune macromolecules are absorbed into the bloodstream [41]. Therefore, this special function of claudin-4 localization is necessary to allow for the proper uptake of immune molecules by the piglet, and subsequent relocalization to the lateral surface may aid in sealing of the paracellular space between adjacent epithelial cells once the immune constituents are absorbed.

Another tight junction protein, tricellulin, serves a special function in the tight junction barrier where more than two epithelial cells meet. Tight junction strands between two adjacent epithelial cells typically associated laterally to pair with another tight junction strand, forming bicellular tight junctions between the two cells [43]. However, these bicellular tight junctions are not continuous at tricellular epithelial cell contacts and have therefore been described as tricellular tight junction proteins. While traditional tight junction proteins, such as occludin and claudins, are found in both bicellular and tricellular tight junctions, tricellulin is concentrated to the tricellular tight junction and its knockdown in the EpH4 cell line of immortalized mouse mammary gland epithelium resulted in altered organization of bicellular tight junction proteins [20]. Additionally, tricellulin has been shown to not affect the permeability for ions while forming a barrier to macromolecules in tricellulin-transfected MDCK II cells overexpressing tricellulin in the tricellular tight junctions [44]. Overall, it is crucial to consider all components and special functions of tight junction proteins when studying tight junction structure. Furthermore, the understanding of these tight junction special functions may be crucial to restoring barrier function following injury.

## 3. Acute Mechanisms of Repair in Injured Intestinal Epithelium

When the monolayer of intestinal epithelium is injured, such as that which occurs during ischemia/reperfusion injury or exposure to pathogenic microbes such as rotavirus [45,46], detachment of the epithelium from the basement membrane and separation of adjacent epithelial cells from one another due to dysregulation and loss of tight junctional proteins occurs. Furthermore, the loss of polarity-establishing tight junctional proteins results in the loss of cell polarity, which abolishes apical and basolateral positioning of localized molecules such as ion channels/transporters, resulting in their mislocalization [47]. When homeostatic positioning of ion channels and transporters is lost, this can subsequently lead to the dysregulation of a multitude of cellular functions including water absorption/secretion, intracellular and organelle pH, and nutrient absorption.

Once the cause of intestinal injury is resolved, such as restoration of blood flow in ischemic injury, rapid mechanisms of intestinal mucosal repair take place in a well-orchestrated series of reparative events. Initially, small intestinal villi contract via the contraction of myofibroblasts adjacent to the epithelial basement membrane and centrally along the central lacteal. Villus contraction is characterized histologically by a quantitatively diminished villus height [48] and occurs in response to mediators such as PGE_2_ [49]. Villus contraction results in reduction of the denuded surface area that remains to be covered by epithelial cells. Simultaneously, restitution of epithelial cells shouldering the site of injury occurs to cover the denuded area [50]. These cells depolarize to disassemble microvilli, allowing for subsequent lamellipodia-driven movement via actin–myosin treadmilling, while maintaining transient attachment to the basement membrane through integrins [11]. Although the underlying intestinal layers may not appear exposed to luminal contents since the mucosa is no longer denuded, the unrestricted permeability pathway via poorly formed tight junctions allows for microorganisms and macromolecules to cross the epithelial barrier. In order for the tight junction barrier and cell polarity to be restored, tight junction proteins internalized during injury, such as the endocytosis of occludin that accompanies anoxic injury in Caco-2 cells, must be reinserted back into the membrane via recycling endosomes [51,52,53]. Ultimately, following these acute repair mechanisms, crypt cells can proliferate and differentiate to restore the proper number of epithelial cells to the monolayer in order to regain full homeostatic function.

## 4. Regulation of Tight Junctions via Ion Channels/Transporters

Closure of the tight junction after acute intestinal injury is paramount in restoring barrier function and returning to homeostatic functioning. Tight junction proteins can be regulated by many factors, including cytokines, growth factors, and nutrients. For example, transport of glucose by SGLT1 has been shown to result in the physiological opening of tight junctions in an NHE3-dependent mechanism [54]. Alternatively, ion channel/transporters, including proteins from the Na^+^/H^+^ exchanger (NHE) family as well as chloride channel protein 2 (ClC-2) have also been shown to regulate tight junction proteins, specifically after intestinal ischemic injury [55,56,57]. This review will examine the reparative role of these transport proteins specifically related to restoration of junctions.

### 4.1. NHE2 and Intestinal Repair

The gastrointestinal epithelium is home to many ion transporters that are collectively responsible for regulating homeostatic cell functions, including the regulation of nutrient absorption, cytosolic and organelle pH, water absorption and secretion, and cell volume [58]. One major family of ion transporters in the human GI tract is the *SLC9* gene family, also known as the NHE family. NHE isoforms belonging to the *SLC9A* gene subgroup (*SLC9A1-9*) can be either plasmalemmal or intracellular, depending on the isoform and tissue location within the gut [59]. Additionally, the Na^+^/H^+^ exchanger 5 (NHE5) is the only isoform for which expression has not been shown in the gastrointestinal tract [60]. These proteins are responsible for the electroneutral antiport of Na^+^ into intestinal epithelium in exchange for H^+^ secreted from the cell to maintain cellular pH and volume.

An additional function of NHEs that continues to be explored is the link between NHEs and the tight junction. One mechanism that links NHEs to the tight junction is through binding to the actin cytoskeleton. Specifically, Na^+^/H^+^ exchanger 3 (NHE3) has been shown to bind directly to the actin cytoskeleton and indirectly through various binding partners, including ezrin [61,62]. The ezrin protein is known to link the plasma membrane to the cytoskeleton in its active, phosphorylated conformation through binding to actin with its C-terminal region [63,64,65]. This interaction with the cytoskeleton has been shown to regulate plasma membrane tension, which is involved in motility and endocytosis [66]. Ezrin links the cytoskeleton to the plasma membrane through binding of its N-terminal region to either membrane lipids or cytoplasmic regions of transmembrane proteins, including NHE3 [66,67]. Thus, by linking transmembrane proteins such as NHE3 to the cytoskeleton, there is an indirect link between transmembrane proteins and tight junction proteins.

Of the NHE isoforms that have been described in the gut, Na^+^/H^+^ exchanger 2 (NHE2) is one of the least described NHEs in regards to its homeostatic and pathophysiologic functionality. However, NHE2 has been linked to paracellular barrier function and tight junction regulation during the recovery of injured intestinal epithelium [55,56]. In both porcine and murine models of intestinal ischemic injury, NHE2, rather than NHE1 or NHE3, appears to be the primary NHE responsible for regulating tight junction proteins during the recovery of ischemia-injured intestines [55,56]. During ex vivo recovery of porcine intestinal ischemia, selective pharmacologic inhibition of NHE2 enhanced epithelial recovery, as evidenced by significant elevations in transepithelial electrical resistance (TER) while inhibition of NHE1 or NHE3 did not elicit a recovery response [56]. In the same study, this NHE2-specific inhibitory effect on recovery was independent of epithelial restitution, and NHE2 was shown to co-immunoprecipitate with ezrin/radixin/moesin (ERM)-binding phosphoprotein 50 (EBP50), also known as NHE regulatory factor 1 (NHERF1), in ischemia-injured porcine ileum. These data suggest that NHE2 regulates restoration of the tight junction barrier during recovery of intestinal ischemia and is potentially linked to the actin cytoskeleton through binding partners (Figure 1). Although NHE2 is also implicated in the in vivo recovery of murine intestinal ischemia, the genetic knockout of NHE2 in the murine model has the inverse effect when compared to pharmacologic inhibition of NHE2 in the porcine model of intestinal ischemia [55]. Specifically, NHE2 null mice exhibit increased blood-to-lumen ^3^H-mannitol flux at 1.5 and 3 hours after ischemic injury as well as a change in localization of occludin and claudin-1 from the membrane to the cytosol when compared to wild-type mice [55]. Additionally, epithelial restitution after intestinal ischemia was unaffected by the absence of NHE2 in this model. It is important to note that pharmacologic inhibition or genetic knockout of NHE2 may affect intracellular pH (pH_i_) since NHEs are known to contribute to pH_i_ changes, and these potential pH_i_ changes can affect charge selectivity of the paracellular pathway [68,69]. However, further studies will be required to determine if NHE-mediated changes in pH_i_ are linked to alterations in the tight junction. Together, this information suggests that NHE2 regulates acute recovery after intestinal ischemic injury in a tight junction-dependent manner, whereas its absence delays restoration of tight junction barrier function.

### 4.2. ClC-2 and Intestinal Repair

Chloride secretion from intestinal epithelium into the lumen is crucial for homeostatic water absorption/secretion via maintaining an osmotic balance with luminal accumulation of both chloride and sodium ions. This subsequently allows for proper mucosal hydration of the epithelial layer, which protects the lumen as food passes through the intestine [70]. The primary protein responsible for chloride transport into the intestinal lumen is the apically located cystic fibrosis transmembrane receptor (CFTR) [71]. However, another contributor to transepithelial chloride transport within intestinal epithelium is the voltage-gated ClC-2 protein, one of nine mammalian proteins belonging to the chloride channel (ClC) protein family [72]. ClC-2 has been shown to localize in the plasma membrane at tight junction complexes within mouse intestinal epithelium [73] or has plasmalemmal basolateral localization within guinea pig colons [74], suggesting species- or tissue-specific localization of ClC-2.

In addition to its role in transepithelial chloride transport, ClC-2 has been shown to regulate intestinal tight junction barrier function in various injury models. After porcine intestinal ischemic injury, stimulation of ClC-2 with the ClC-2 agonist lubiprostone during ex vivo recovery on Ussing chambers resulted in marked increases in TER and reduced mucosal-to-serosal mannitol flux [57]. Contrasting the effect of ClC-2 stimulation with lubiprostone, the genetic absence of ClC-2 in a murine model of intestinal ischemia resulted in significant increases in blood-to-lumen mannitol clearance while also reducing expression of membrane-bound occludin and claudin-1 after up to 3 hours of in vivo recovery [75]. In this murine model, occludin co-localized with ClC-2 after co-immunoprecipitation studies, and its localization to the tight junction region was diffuse in ClC-2 null mice after up to 3 hours of recovery [75]. Additionally, in a murine model of dextran sulfate sodium (DSS)-induced colitis, the absence of ClC-2 increased disease severity, as measured through significant losses in body weight and significant increases in disease activity index [76]. ClC-2 null mice treated with DSS also demonstrated significantly increased expression of claudin-2 and reduced occludin expression in the same study. Interestingly, a recent in vitro study established Caco-2 cells overexpressing ClC-2 (Caco-2^ClCN2^), and this ClC-2 overexpression resulted in a decrease of the pore-forming claudin-2 protein while maintaining claudin-1 and claudin-4 protein levels to that of control cells [77]. As an aside, although cell volume and pH_i_ is partially regulated by ClC-2 and thus the genetic knockout of ClC-2 can affect these intracellular factors, studies will be needed to determine if ClC-2-mediated changes in these intracellular factors have an effect on the tight junction [78,79]. Based on these studies, there appears to be a mechanistic link between ClC-2 and the regulation of membrane claudin expression, but further studies will need to be carried out to determine how ClC-2 plays a role in claudin expression patterns. Nonetheless, current data suggest the critical role of ClC-2 in barrier function during recovery from epithelial injury while also reinforcing the link between ClC-2 and the tight junction barrier.

The link between ClC-2 and the tight junction was initially shown to exist through intracellular caveolar trafficking of occludin via interaction with both caveolin-1 and the small GTPase Rab5 in a cell line derived from human intestinal Caco-2 cells (Figure 2) [80]. This connection between ClC-2, occludin, and caveolin-1 was further supported in vivo with a model of DSS-induced colitis. ClC-2 null mice treated with DSS had significantly displaced occludin/caveolin-1 densitometry readings toward high-density, detergent-soluble fractions of sucrose density gradient-based fractions when compared to wild-type mice treated with DSS [76]. These data suggest that after DSS-induced colitis, occludin and caveolin-1 are strongly associated in the cytosol of mice lacking ClC-2 but not in mice normally expressing ClC-2. In tandem, overexpression of ClC-2 in Caco-2^ClCN2^ cells was reported to not only exhibit enhanced tight junction barrier function through significant increases in TER and reductions in apical-to-basal inulin flux, but this ClC-2 overexpression further connected ClC-2 to caveolin-1 and caveolar trafficking of occludin [77]. Specifically, ClC-2 overexpression in Caco-2^ClCN2^ cells exhibited both significantly increased occludin protein and reduced endocytosis of occludin when compared to control cells while simultaneously diminishing both caveolin-1 protein and caveolae assembly [77]. Furthermore, this study reported that selective inhibition of ClC-2 lead to both reduced occludin protein and increased caveolin-1 protein. Taken together, there is strong evidence from both in vitro and in vivo models that links ClC-2 to the tight junction protein occludin and its regulation by caveolar trafficking. Based on the presented evidence, it is believed that ClC-2 facilitates the shuttling of endocytosed tight junction proteins back to the apical–lateral membrane to repair injured tight junctions. However, further mechanistic studies are required out to determine the precise mechanisms of these events.

## 5. Conclusions

Injury of intestinal epithelium affects both the epithelial cells and the junctional structures that link them. A great deal of attention has been focused on mechanisms of epithelial restitution, but a lesser level of attention has been paid to the reassembly of tight junctions within repairing epithelium. This intriguing process appears to be intimately associated with ion channels, which in the case of NHE2 and ClC-2, is facilitated by a close association with tight junction regulatory proteins. With ClC-2 in particular, the mechanism of ion channel-facilitated tight junction reassembly has been linked to endosomal recycling of tight junction proteins, with evidence of restoration of the positioning of tight junction integral membrane proteins during the reparative process, and increased membrane expression of sealing claudins with cellular over-expression of ClC-2. However, how precisely ion channels interact with structures such as endosomes, and how this facilitates insertion of sealing tight junction proteins at the repairing tight junction will require further study. Nonetheless, it does appear that ion channels such as NHE2 and ClC-2 have a greater cellular function than ion transport alone. It is conceivable that the transport of select ions accompanies a structural change that sets off a series of signaling events associated with tight junction reassembly, but this will require additional mechanistic work. Ultimately, further studies to uncover the relationship between ion channels and reassembly of tight junctions has the potential to lead to novel therapeutic targets for patients with increased intestinal paracellular permeability.

## Figures and Tables

**Figure 1 ijms-21-00972-f001:**
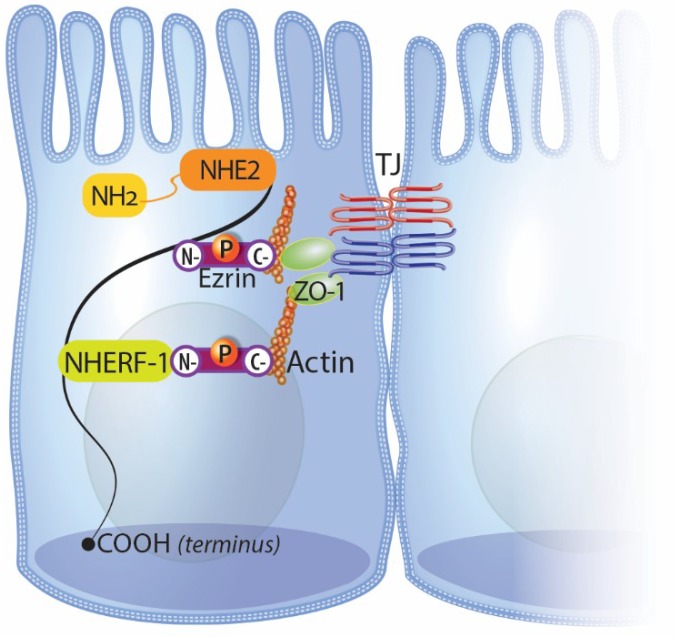
Schematic representation of the potential link of NHE2 to the actin cytoskeleton and subsequently the tight junction through binding partners. The primary candidate protein linking NHE2 to the actin cytoskeleton is phosphorylated ezrin. Based on information known about NHE3 and data from NHE2 in vivo studies, NHE2 may bind directly to ezrin or indirectly through additional binding partners, including NHERF1/EBP50.

**Figure 2 ijms-21-00972-f002:**
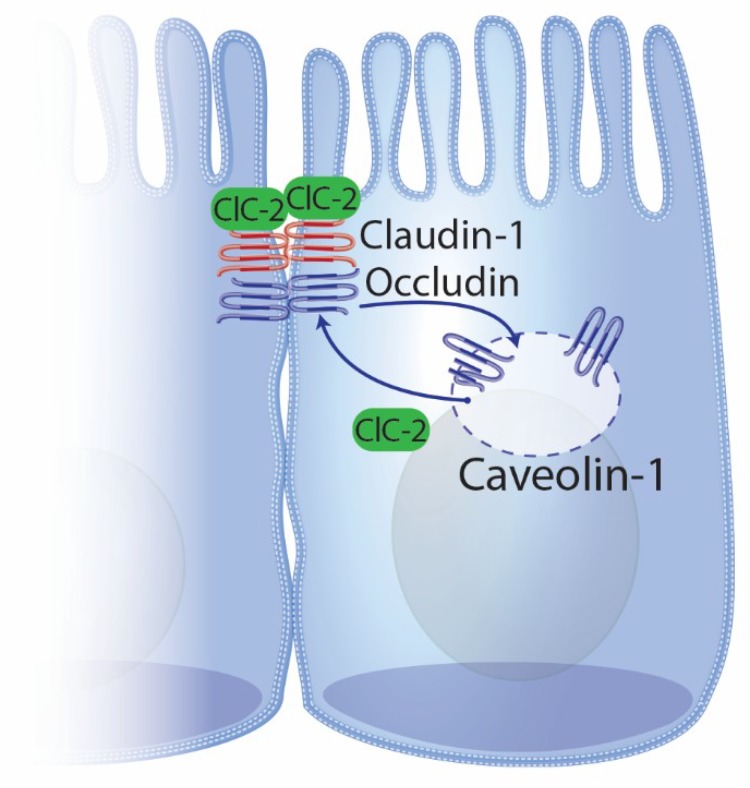
Schematic representation of the link between ClC-2 and caveolar trafficking of occludin. This schematic is a slight modification from a previously published figure [81] to more closely associate ClC-2 to both the tight junction complex and to caveolin-1-associated endocytosis and recycling of tight junction proteins such as occludin. Note that the representation of ClC-2 at the tight junction complex is not exclusively apical in localization, which leaves room for ClC-2 to be more closely associated with occludin in the membrane.

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
