# Peer review of "The Integral Role of Tight Junction Proteins in the Repair of Injured Intestinal Epithelium"

_ijms, 2020, doi:10.3390/ijms21030972_

Round 1
Reviewer 1 Report
The author reviews the tight junction biology, acute mechanisms of repair in intestinal epithelium, and the regulation of tight junctions through ion channels/transporters. It is an interesting and important topic to discuss about tight junction activity during epithelium injury repair, and this manuscript represents a clear outline of statements with extensive reference from latest publications. But there are still some aspects to address before publication:
(1) There are several statements lacking references. For example, page 1, line 35; page 2, line 77; page 3, line 138; page 4, line 143, 164, 166, 177.
(2) Page 2, paragraph 2: the introduction of sealing claudins and pore-forming claudins need to reorganized to become clear to readers. "Interestingly, sealing claudins can reduce paracellular permeability through displacement of pore-forming claudins", this sentence needs to be explained better, maybe expanded the explanation.
(3) Page 6, paragraph 1: the role of ClC-2 for tight junction trafficking and junctional structure formation needs to be better defined. Is that ClC-2 strengthens the rescue of tight junction proteins from endocytosis? If so, make it clear.
(4) The conclusion paragraph needs rewriting and expansion. It is expected that the authors provide a good summary of mechanisms from different kinds of related literature, and their own perspective of future direction. However, I don't see a clear summary of relationship among injured epithelium repair, transporter role and tight junction modeling. I also expect the author to state needed research to be carried out to depict the uncertainties about above relationships, and give envisions for the future directions of this field.
Reviewer 2 Report
The paper entitled “Integral role of tight junction proteins in repair of injured intestinal epithelium” by Slifer and Blikslager reviews the repair of tight junctions after the acute injury in the intestine and the roles of NHE3 and ClC-2 in the repair. They provide an important topic in the acute regulation of tight junctions and the roles of transporters in this regulation. However, there are several aspects to address before publication as listed below.
(1) The authors state that the absorption and the barrier to the luminal contents in the intestine are primarily maintained and regulated by tight junctions. However, a transcellular pathway also paly an important role in the absorption and the barrier in the intestine. The authors should mention it.
(2) It is stated that injury of the intestinal barrier occurs through tight junction protein dysregulation in the various pathological conditions. The references of this statement should be included. Also, the injury of the intestinal barrier also occurs through the cytotoxicity in the intestinal epithelial cells. The authors should mention it.
(3) In section 1, it is noted that ”Inability to rapidly repair the tight junctions in order to restore epithelial barrier function is detrimental to the patient, as it can result in various pathologies, including nutrient malabsorption, improper water absorption/secretion, sepsis, and multiple organ dysfunction [5-7]”. However, the references [5-7] are the studies about claudin KO mice and the effects of sepsis on the intestinal barrier. The authors should indicate appropriate references.
(4) In section 2, the authors state that ”Alternatively, increased expression of pore-forming proteins (including claudins-2, -10a/-10b, -15, -16, -17) decreases the selectivity of luminal contents that can pass between epithelial cells”. It is not clear what this statement means. Also, the references should be included.
(5) It is mentioned that “Additionally, its protein expression was detected throughout the crypt-villus axis of the small intestine but was only detected in undifferentiated crypt cells of human colonic tissue [32]”. However, the expression pattern of claudin-2 in the small intestine is different among species. In fact, later in the paragraph, the authors also refer to the restricted expression of claudin-2 in the crypt. The authors should revise the manuscript.
(6) The authors describe the change of claudin-4 localization after the birth and the role of tricellulin in section 2.1. However, relation and importance of the section 2.1 in this review is not clear.
(7) In section 3, the authors state that “In order for the tight junction barrier and cell polarity to be restored, tight junction proteins internalized during injury must be reinserted back into the membrane via recycling endosomes [48, 49]”. The examples that show the recycling of tight junction proteins in the recovery from acute injury should be indicated.
(8) In section 4.1, the authors describe the regulation of tight junctions by NHE2 via the actin, which is involved in the recovery of tight junctions after the ischemia in the intestine. If NHE2 regulates tight junctions via the actin, the inhibition of NHE2 or the knockout of NHE2 is expected to cause difference in the actin during the recovery from the ischemia. Are there any studies that show the effect of the inhibition or knockout of NHE2 on the actin? Also, it is likely that the changes in the pH and/or osmolality by the inhibition or knockout of NHE2 also affect the function of tight junctions. The authors should refer to this possibility.
(9) In section 4.2, the authors mention that ClC-2 is involved in the endocytosis of tight junction proteins via caveolin-1 and regulates the barrier function of tight junctions. However, the change in the expression pattern of claudins is also important in the regulation of the tight junction barrier especially in the cases like DSS-induced colitis and ClC-2 overexpression. In fact, the authors refer to the increase of claudin-2 expression in DSS-induced colitis in the ClC-2 KO mouse. The authors should also discuss the role of ClC-2 in regulation of the claudin expression pattern. Also, it is likely that the changes in the intracellular chloride concentration, pH and/or cell volume by the inhibition or knockout of ClC-2 also affect the function of tight junctions. The authors should refer to this possibility.
Reviewer 3 Report
This review article is written well and deserve publishing on journal. Thank you for your submitting your article.
Author Response
We thank the reviewer for the encouraging comments
Round 2
Reviewer 2 Report
The manuscript has been much improved. I would agree to publish with the current edition.
Author Response
We are grateful to the reviewer for the positive comments, and have accepted all changes, checked spelling, and checked grammar on the revised version